# A fog-assisted group-based truth discovery framework over mobile crowdsensing data streams

**Bayan Hashr Saeed Alamri**[1,2], **Muhammad Mostafa Monowar**[2], **Suhair Alshehri**[2], **Mohammad Haseeb Zafar**[3*]

**1** Faculty of Cyber Security and Forensic Computing, Prince Mugrin University, Al-Madinah, Saudi Arabia,
**2** Faculty of Computing and Information Technology, King Abdulaziz University, Jeddah, Saudi Arabia,
**3** Cardiff School of Technologies, Cardiff Metropolitan University, Cardiff, United Kingdom

* mhzafar@cardiffmet.ac.uk;m.h.zafar@ieee.org

**Data availability statement:** The data is available in files within the Figshare repository https://doi.org/10.6084/m9.figshare.26251022.

## Abstract

With the proliferation of mobile crowdsensing (MCS) and crowdsourcing, new challenges are emerging every day. Although crowdsensing has become a popular sensing paradigm to aggregate sensor readings from a variety of sources, data inconsistency has arisen as a serious challenge. Truth discovery (TD) has been developed as an effective method for reducing data inconsistency and as a validity assessment for conflicting data from various sources. In addition, MCS applications and services are moving beyond a single individual participant to community groups and are influenced by group behavior. To address these challenges in this paper, we propose a novel Fog-assisted Group-based Truth Discovery Framework over MCS Data Streams, an efficient TD system for real-time applications. Specifically, we first initialized the weights for the weight update process in TD with the participants' credibility level. Then, we developed a novel Two-layer Group-based Truth Discovery (TGTD) mechanism in which the first layer estimates the truth of the group's members and the second layer estimates the aggregated truth for the groups. We have conducted extensive experiments over synthetic and real-world datasets to prove the effectiveness and efficiency of our framework. The results indicate that TGTD achieves superior truth discovery accuracy compared to current streaming truth discovery approaches, while maintaining a reasonable running time. The organization of the streaming process within the fog architecture simulation is identified as an area for further investigation and future work.

## Introduction

In the mobile crowdsensing (MCS) paradigm, data collected by participants are not always accurate or reliable due to background noise, sensor and hardware quality, a lack of effort, and insufficient skill. Moreover, the sensor readings of participants on the same tasks may differ [1–3]. Therefore, while sensor readings are highly valuable, it remains a research challenge to extract truthful data from the conflicting, heterogeneous, and noisy readings reported

**Funding:** The author(s) received no specific funding for this work.

**Competing interests:** The authors have declared that no competing interests exist.

by participants. Decisions based on untruthful data can cause serious damage. For example, serious consequences may result from patients' medical records being scattered across different hospitals, and wrong diagnoses can occur when based on incorrect measurements [4]. Furthermore, scientific discovery can be guided in the wrong direction due to faulty data [5]. Therefore, it is crucial to extract the most trusted and reliable results from conflicting sources. Significant improvements in the accuracy of data aggregation have been brought by researchers on the quality of ground truth [6–9]. Hence, to discover truthful information from unreliable data, a possible solution could be truth discovery (TD) [10–12], which has been widely studied recently and applied in many MCS applications [13–15]. The TD goal is to estimate participants' and workers' data quality and infer reliable aggregated information through quality-aware data aggregation [16,17]. The TD process aims to estimate the truth, which is the closest value to the true value of the task based on participants' weights as a reliability input. The main principle of the TD method is that participants will be given a high weight if they contribute sensor readings closer to the ground truth. In addition, their sensor readings will be counted more in the aggregation process. Several studies have focused on TD techniques and variety of TD approaches have been proposed to calculate participants' weights and aggregated results based on TD principles [16,17]. It is intuitive to trust reliable participants more when deriving the truth, and the naïve approach that regards all participants as reliable and equal in the aggregation may fail to infer reliable results. In most studies on TD, the researchers have assumed that most participants and workers are reliable [8,11,18–23] though this assumption is not practical in most MCS scenarios. Especially under circumstances where the ground truth is unknown. They either rely on the observation that the majority of users contribute reliable data even in the absence of ground truth [8,11]. Some methods initialize a random ground truth at the beginning of the TD process [18], assuming the worker weight to be known [19,23], or assign random weights [20–22] at the start. Formally, for most of the previous studies, as long as most of the participants honestly contribute sensing data, TD can generate an effectively estimated truth [3,6,8,18–32]. What happens, however, if the misbehaving participants are dominant at first? Specifically, the quality estimation in these studies starts with zero knowledge of participants' reliability. If the misbehaving participants dominate at first, then the quality estimation will be likely to be inaccurate, and they will incorrectly consider these participants as reliable once, accordingly generating false truth estimations [8]. Furthermore, in most truth discovery scenarios data are collected in a streaming manner, where it is reported from multiple sources sequentially, such as traffic monitoring applications, flight data, and weather forecast information [33]. Consequently, it is impractical to wait for all the sensor readings to be collected to estimate the truth and source reliability. As this framework focuses on real-time MCS applications, it resorts to the incremental conflict resolution (iCRH) algorithm [16], which deals with streaming data [23,34]. In addition, MCS applications and services are moving beyond a single individual participant to community groups and are influenced by group behavior [35]. MCS grouping exists to support real-world group activities (e.g., meetings, parties, etc….). In addition, grouping is an important phase of the design space for MCS systems such as management, economics [36], social networks, and social influence phenomena [37,38].

In this work, what our framework can accomplish is that even if the majority of participants are unreliable, we can still generate accurate truth estimation results. This is because, by initializing the weights in the TD process with the participants' credibility level, our estimation approach tends to let the high-weight participants contribute more to the TD process, which allows the truth estimation result to be accurately generated. In addition, we adopt the Fog Node (FN) architecture to minimize the overhead on the participants' side. Such that the FN are located in different geographical locations and provide fog computing services for

both the participants and the sensing platform [39]. The FN communicates with the participants, forwards their sensor readings to the platform, and helps in the TD process along with the sensing platform.

In this paper, we address four challenges when developing the Fog-assisted Group-based Truth Discovery Framework over MCS Data Streams, an efficient TD system for real-time applications. First, participants may be lazy and selfish when collecting sensor readings with their resource-constrained devices. Hence, they may reduce their sensing efforts, such as by reducing their resources, time, and attention in the sensing tasks, which will significantly impair the aggregation [40]. Accordingly, the TD algorithm needs to start the process with credible participants' weights. Second, sensor readings are collected as streams rather than as static data in real-time applications; hence, each participant submits sensor readings at regular intervals. This challenge requires a streaming TD algorithm [16,41]. Third, we address the shortcomings of existing group-based TD in MCS. Therefore, we present a novel Two-layer Group-based Truth Discovery (TGTD) mechanism to calculate TD for group-based activities in MCS. Finally, for large-scale deployment of real-time applications, the TD scheme should be efficient, even if the number of participants grows dramatically. With the increasing complexity and scale, there is a need to enhance MCS with a fog-computing paradigm to reduce computation complexity and communication overhead. Each FN manages a group of participants and works as an intermediate between participants and the sensing platform.

To the best of our knowledge, this work is the first that investigates group-based TD. Moreover, existing works start the TD process to estimate the data quality with zero knowledge about the ground truth and participants' weight or uniform initialization of user weight; therefore, if the misbehaving participants dominate at the start, it is likely to generate false truth estimation and classify these participants as reliable. To the best of our knowledge, this work is the first to start the TD process with a credible participant's weight. Finally, existing works failed to achieve efficient TD for a large population of participants.

In particular, the main contribution of this work is A Fog-assisted Group-based Truth Discovery Framework over MCS Data Streams. Overall, our contributions can be summarized below:

- We develop a novel TD method for crowdsensing data streams where we initialized the weights in TD with participants' credibility level. Then we formulate the credibility function based on the participants' readiness, commitment, and the device's ability to perform the sensing tasks. This method guarantees the accuracy of the data as participants with a high credibility level are likely to provide accurate data.
- We develop a novel fog-assisted Two-layer Group-based Truth Discovery (TGTD) mechanism. The first layer estimates the truth of the group's members and the second layer estimates the aggregated truth for the groups in the area of interest. Additionally, it offloads the process to the FN deployed in a different geographic location, to minimize the overhead on the participants' side.
- We conduct extensive simulations using synthetic datasets and different real-world datasets and verify that TGTD surpasses other streaming TD methods in terms of TD accuracy, while maintaining reasonable computation time.

The remainder of this article is organized into the following sections: Related work section reviews related work in TD schemes in MCS. Preliminaries section states our system model, design goals, and brief preliminaries of truth discovery. A Fog-assisted Group-based Truth Discovery Framework over MCS Data Streams and its underlying mechanisms are introduced

in Proposed framework section. In Performance evaluation section, the framework is evaluated by conducting various experiments and simulations. Finally, we conclude the article in the conclusion.

## Related work

Many studies have directed their attention toward exploring the truth discovery problem, which is an effective technique for MCS quality-aware data aggregation from heterogeneous sources [16,17,42]. Li et al. [17] propose a general truth discovery framework, which has been utilized in many other studies. They propose Conflict Resolution on a Heterogeneous (CRH) data approach, which iterative conducts truth aggregation and weight estimation until convergence. Furthermore, in [16] a general truth discovery framework for single, heterogeneous, and steam data types is proposed. Li et al. [42] proposed confidence-aware truth discovery (CATD) to automatically infer truths from conflicting data with long-tail phenomenon.

Thereupon, multiple researchers have adopted the same concept of CRH to find truth discovery from multiple observations to qualify sensor data and reward participants based on their contribution to the truth. The data quality in these studies is based on the deviation between reported sensor data and the ground truth [8,18–20,22–24,26–30]. Yang et al. [8] propose a quality estimation model through unsupervised learning. They adopt the idea of clustering for ground truth estimation and measuring the data quality of each participant. In this model, data quality is based on the deviation between trustworthy data and the ground truth. It achieves better performance in terms of quality estimation compared with other heuristic models. On the other hand, it assumes that every mobile device has equal sensing capabilities; however, the participants' behaviors are uncertain, and their sensing devices are heterogeneous. Moreover, it assumes that quality data providers dominate at the start of the truth estimation model, which is an impractical assumption in the real-world MCS.

Others address privacy concerns during the truth discovery process [19,20,23–26,28–30,39,43] where they either adopt the Paillier cryptosystem [18,23,26,27,43], anonymity [23,30,39], or differential privacy [19,20,22,30,34,44] to protect customers' and participants' sensitive information and the estimated weight. The authors in [39] propose a fog-assisted data collection scheme for MCS. This scheme utilizes a session key agreement mechanism for the MCS data collection environment. It achieves data anonymity and accuracy without relying on TTP. Liu et al. [23] tackle the dropout of participants in the MCS system by proposing robust and scalable TD in real-time MCS applications. It processes sensing heterogeneous data streams based on the iCRH. This scheme achieves highly efficient computation and enough accurate truthful information. In [30], the authors tackle the problem of truth discovery protocols that impose heavy computation and communication overhead. They propose the PerturbTD protocol to reduce participant overhead when they are sharing their weight during the truth discovery process. PerturbTD reduces the overhead on the participant by shifting the truth discovery process to be performed on two cloud servers. Gao et al. [43] propose a novel and efficient location privacy-preserving truth discovery (LoPPTD) mechanism. It investigates location-preserving truth discovery for MCS. It divides the interested area into grids and exploits super-increasing sequences. Then, participants structure their sensory reading into a report that is uploaded to the FN to apply the truth discovery process. Zhao et al. [27] propose PRICE, a privacy-preserving and reliability-aware real-time incentive system in MCS. Furthermore, they design a two-layer stream truth discovery model. This model data stream tackles the single-time slice of the failure STOF problem by adding one round truth discovery layer, in which the first layer processes the sensing data stream and the second layer processes estimated ground truth from the first layer. The problem of reliable aggregate results

and protecting participants' information in truth discovery is addressed in [26]. It adopts the truth discovery method CRH and assigns participants' weights based on information quality. Two schemes are proposed that use a two-layer randomized response mechanism and a Gaussian noise mechanism. These schemes adopt the truth discovery method, CRH, and assign participants' weights based on information quality. Hence, the aggregated data do not deviate much from the true value. The authors in [44] propose a LightPrivacy scheme to balance participants' personalized privacy and task data practicability in MCS. Chen et al. [22] propose a novel robust privacy truth discovery scheme called RPPTD in which differential privacy allows the operation server to obtain the truth without leaking the participants' privacy. This scheme adopts CRH as a truth discovery process that includes weight update and truth update processes. RPPTD achieves robustness and eliminates single-point failures without the need for trusted third-party servers. The location privacy truth discovery MCS system is addressed in [20], where the authors propose location obfuscation truth estimation based on differential privacy. They model location privacy as an optimization problem that is solved via linear programming to minimize global truth estimation deviation differential privacy constraints. In [19], A novel framework called Truth discovEry via probabilistic eStimation mall under rigorous Local differential privAcy (TESLA) is introduced. In this framework, the privacy-protected noise weakly negatively affects the weight estimation and true aggregation. In addition, TESLA includes a probabilistic weight mechanism for determining a more accurate weight for each participant. Moreover, based on the fused value, the sensing platform computes the truth discovery process while adopting CRH and utilizes a probabilistic weight mechanism as weight estimation for truth discovery. Although there is a need to consider stream-sensing data tasks, TESLA achieves high effectiveness and efficiency. However, these systems impose an overhead on participants and the sensing platform, which discourages them from engaging in the truth discovery process.

Some works improve the efficiency and performance of truth discovery systems by developing their systems under fog or edge computation [22–25,32,34,39,44–48]. Accordingly, they shift the aggregation of sensor data and the truth discovery process into the external node. Xu et al. [45] tackle the problem of false event reports generated in MCS systems. They design a fog-assisted crowdsensing architecture for vehicular applications. This system solves the trust assessment issue by converting it into a maximum likelihood estimation problem. Accordingly, they solve this problem through the expectation-maximization algorithm. The FN in this scheme verifies data trustworthiness and filters and then uploads local traffic conditions to service providers and cloud servers. The authors in [32] address truth discovery in real-time applications with large number of participants. They propose a fog-aided privacy-preserving truth discovery framework that is secure and efficient in handling real-time applications with a large group of participants. In addition, they designed a unique secure aggregation protocol, SecAgg, which can securely and efficiently aggregate inputs from workers in smaller groups. This framework adopts cloud-fog computing architecture to divide the complete worker group into many smaller ones. In [25], edge-assisted truth discovery for large-scale MCS by utilizing CRH to estimate the truth for both the deep cloud and edge cloud. In addition, they propose an incentive mechanism consisting of truth discovery and reverse auction stages. It shows superiority in terms of estimation precision, but it incurs high computation and communication overhead. Zhang et al. [21] tackle truth discovery for stable and moving participant MCS applications. They update the reliability and the ground truth and filter out false data before sending them to the cloud. Furthermore, keeping the computational costs and communication overheads minimal. In these schemes, the fog nodes play an intermediary role between the participants and the sensing platform. They aggregate the sensory data and transmit the aggregated results to the platform. Although they achieve high

efficiency and practicality, it is difficult to detect outliers in these schemes. Moreover, these schemes are designed under the two-server settings.

Equally important, multiple studies focus on encouraging participants to engage in improving truth discovery accuracy [18,23,25,44,47] by developing an incentive mechanism to make sure enough trusted sensor data are used in the truth discovery process. The problem of estimated truth discovery on an edge cloud and the incentive for participants to contribute to the truth discovery process is addressed in [25]. They propose edge-assisted truth discovery for large-scale MCS by utilizing CRH to estimate the truth for both the deep cloud and edge cloud. In addition, they propose an incentive mechanism consisting of truth discovery and reverse auction stages. A privacy incentive mechanism based on truth discovery called PAID is proposed in [18]. It sets task constraints such as spatial, temporal, and the type of sensing data to remove untrustworthy participants' sensor data that do not satisfy these constraints. Then, it calculates the truth discovery based on the remaining qualified participants. In PAID, the servers get the aggregation result and adopt CRH to iteratively calculate the ground truth. Moreover, the data quality is calculated based on the participant's weight.

Since most of these works are designed for one-time truth discovery, an iCRH is introduced in which the heterogeneous data streams are processed in each time slot [22,23,32,34]. Similarly, [41] introduced centralized S-CATD for streaming crowdsourcing data. Liu et al. [23] process sensing heterogeneous data streams based on iCRH. The participants report their masked data and truth level to the server. Then, the server utilizes secure summation aggregation to learn the sum of participants' weights and the sum of participants' sensing data to compute the truth of the sensing data. This scheme achieves highly efficient computation and enough accurate truthful information. Chen et al. [22] address the robustness of the truth discovery framework with a single server model. They propose a novel robust privacy truth discovery scheme called RPPTD in which differential privacy allows the operation server to obtain the truth without leaking the participants' privacy. This scheme adopts CRH as a truth discovery process that includes weight update and truth update processes. RPPTD achieves robustness and eliminates single-point failures without the need for trusted third-party servers. Wang et al. [34] tackle the truth discovery problem in streaming crowdsourcing tasks and the privacy of the workers. They propose an edge computing-based privacy-preserving truth discovery scheme for streaming crowdsourcing tasks called PrivSTD. It utilizes edge servers to enable workers to estimate local truths and their reliability, based on which the incentive and perturbation mechanisms are developed. It considers correlations among truths over time and the characteristics of participants' reliability. Mukkamala et al. [41] address the challenge of providing reliable and scalable truth discovery on general streaming data, aiming for higher accuracy and lower cost. They introduce both centralized and decentralized streaming schemes tailored for crowdsourcing applications. These schemes leverage CATD [42], incorporating iterative procedures to enhance TD accuracy. The centralized streaming CATD updates participants' weights based on their task performance.

To the best of our knowledge, no studies have been published on group-based truth discovery. Furthermore, existing works failed to achieve efficient truth discovery for the large group of participants. In addition, they start the truth discovery process to estimate the data quality with zero knowledge about the ground truth, participants' weights, or uniform initialization of participants' weights. Hence, if the misbehaving participants dominate at the start, it is likely to generate false truth estimation and classify these participants as reliable. Moreover, the truth discovery process in the prior works iteratively conducted aggregation and weight estimation steps until convergence. The convergence criterion can be a threshold for the change of the aggregated results in two consecutive iterations or a predefined iteration number. As a result, overhead is imposed on the participants' side.

In contrast to prior truth discovery systems, our system focuses on estimating group-based truth discovery. Accordingly, it supports data quality measurement in any group-based MCS application. In addition, it improves the quality of sensor data by estimating the truth while considering only credible participants. Hence, we examine first the participants' credibility and readiness to engage in the sensing task. In the meantime, the overhead on the participants' side is minimized by leveraging fog-based computing, which assists in calculating the truth value and data quality. Our truth discovery system is built on the iCRH approach due to its state-of-the-art efficiency performance on the data stream [22,23,32,34]. In addition, we adopt a non-private version of the iCRH in this work. Table 1 summarizes the characteristics of this framework in contrast to the above truth discovery schemes.

## Preliminaries

In this section, we first present the system model, and design goals, and illustrate the underlying TD algorithm. The main notations used in this paper are outlined in Table 2.

### System model

We consider a crowdsensing scenario, where the sensing platform monitors a phenomenon without knowledge of the ground truth. The architecture of the A Fog-assisted Group-based Truth Discovery Framework over MCS Data Streams mainly involves four entities:

1. **Participant**: Is the mobile device user who collects the sensor data about some sensing tasks and submits them to a near fog node.
2. **Fog-node**: An entity at the edge of the network with computation capabilities. It is responsible for aggregating the sensing data, managing the group of participants, initializing the weights with the participants' credibility level, performing the TD process, and uploading the aggregate result to the sensing platform.
3. **Sensing platform**: This is an MCS platform that assigns sensing tasks to the participant, collects the aggregated result from the fog nod, cooperates with the FN to perform the TD process, and sends the quality result to the requester.
4. **Requester**: Is the end user who publishes the sensing task about some phenomena and requests accurate sensor data about this task.

**Table 1. Comparison with other TD schemes.**

| Scheme | Stream input | Non-iteration | Group-based TD | Source Credibility | Computational complexity | Communication overhead |
|---|---|---|---|---|---|---|
| [8] | No | No | No | No | High | High |
| [23] | Yes | Yes | No | No | Medium | Low |
| [34] | Yes | Yes | No | No | High | Low |
| [32] | Yes | Yes | No | No | High | Low |
| [22] | No | No | No | No | High | Low |
| Our framework | Yes | Yes | Yes | Yes | Medium | Low |

**Table 2. Notations and descriptions.**

| Notations | Descriptions |
| --- | --- |
| $C_i$ | Participant's $i$ credibility level |
| $[k_1, k_2, k_3]$ | Weights factors, specify the importance of each parameter |
| $C_{i,old}$ | Participants' $i$ old credibility level |
| $R_i$ | Participant's $i$ readiness and capability |
| $Y_i$ | Participant's $i$ ability ratio |
| $E$ | Battery level |
| $A$ | Sensors availability |
| $w_i^0$ | Initial weights |
| $x_i$ | Participants $i$ sensor reading |
| $x_{S,i}^t$ | Participants $i$ sensor reading of task $S$ during time slot $t$ |
| $w_i$ | Participant's $i$ weight. |
| $w_i^t$ | Participant's $i$ weight in the time slot $t$ |
| $\alpha$ | Decay rate |
| $t, t\text{-}1$ | Time slot, previous time slot |
| $I, i$ | Participants, individual participant |
| $S$ | Sensing task |
| $F$ | Fog-nod |
| $X_s^t$ | Estimated truth for the task $S$ during time slot $t$ |
| $X_s^*$ | Ground Truth of Task $S$ |
| $G, GZ$ | Group, group size |
| $G^*$ | All the groups |
| $D$ | Distance function |
| $\acute{C}_i$ | Normalized credibility level of participant $i$ |

In this work, the quality of each participant refers to weight, the ability and readiness of the participant to perform a sensing task as credibility, and the truth of each task as ground truth $X_S^*$. In our model, we formalize the problem by assuming that there are $|S|$ sensing tasks, $|I|$ participants, and $|F|$ fog-node. Such that, the number of tasks $\{1,..,|S|\}$, participant $\{1,\dots,|I|\}$, and FN $\{1,..,|F|\}$. In each time slot $t$, each participant collects sensor reading $x_{S,i}^t$, then the estimated truth is generated as $X_S^t$. The participants with the higher weight, $w_i^t$, are more likely to be considered reliable. However, in a realistic scenario starting the TD process to estimate the data quality with zero knowledge about the ground truth and participants' weight, and randomly or uniform initialization of user weight with unknown values, can lead to fault results if unreliable participants dominate at first. Furthermore, it is hard to obtain a more trustworthy sensor reading without knowing the more credible and dependable participants who possess the required attributes.

## Design goals

In this article, we intend to devise a TGTD mechanism, that can provide more accurate ground truth for group sensing activities. Specifically, our mechanism achieves the following twofold design goals.

1. **Accuracy**: The proposed scheme should output a highly accurate estimation. Accordingly, it is measured by the deviation between the estimated results and the real truths.
2. **Efficiency**: The proposed scheme should reach a significantly lower overhead for the participants. Thus, it is measured by the running time of the system to show the scalability and efficiency.

## Truth discovery

As the underlining TD algorithm, we adopt the iCRH [16]. The iCRH estimated the truth for the sensor data, in which the sensor reading for participants arrives in a streaming manner. In each time slot $t$ there are three steps in this process: truth update, distance update, and weight update [23,32]. Algorithm 1 shows the iCRH truth discovery for the data stream in MCS.

**Algorithm 1. Truth discovery [16].**

```
    Input:  Sets of participants' sensor readings xᵗ, decay
rate α
    Output: Estimated truth X*ₛ
1: Initialization:
2: The truth Xᵗₛ = 1/|I| Σ|I|ᵢ₌₁ xₛ,ᵢ
3: while Not convergent do
4:    Estimate weights with Eq 1
5:    Estimate distance with Eq 2
6:    Estimate truths with Eq 4
7: end while
8: return: X*ₛ
```

1. **Truth update**
   This step estimates the truth of the sensing task at time slot $t$ as a weighted average of the participants' sensor reading $x_i^t$, and the latest estimation of the participant's weight in the last time slot $w_i^{t-1}$.

$$X_S^t = \frac{\sum_{i=1}^{|I|} w_i^{t-1} * x_{S,i}^t}{\sum_{i=1}^{|I|} w_i^{t-1}} \tag{1}$$

2. **Distance update**
   In this step, the incremental distance is updated based on the sensor reading, the estimated truth, and the previous distance.

$$D_i^t = D_i^{t-1} \cdot \alpha + D(x_{s,i}^t, X_s^t) \tag{2}$$

   Where $D(.)$ is a distance function to measure the deviation level between the participant $i's$ sensor reading for task $S$ ($x_{S,i}^t$), and the estimated truth for the same task ($X_S^t$). Note that, the distance function is chosen based on the MCS application, the task requester, and the sensor reading type. In this framework, we focus on the continuous data type, where the commonly adopted distance function is the square distance [21,23].

$$D(x,y) = (x_{s,i}^t - X_s^t)^2 \tag{3}$$

   To allow the recent reading to play a more important role in TD, iCRH utilizes the decay rate $\alpha$ [16,21].

3. **Weight update**
   In this step, each participant's weight is updated incrementally based on the distance function of the previous $t-1$ slot. The basic idea is that the smaller the distance between

the participants' sensor readings and the current truth, the higher the weight the participant gets.

$$w_i^t = log\left(\frac{\sum_{i=1}^{|I|} D_i^t}{D_i^t}\right) \tag{4}$$

## Proposed framework

### Overview

This section presents and discusses the Fog-assisted Group-based Truth Discovery framework in detail and its underlying mechanisms. This framework takes the sensor readings from participants in the AoI, quantifies the participants' credibility, and then initializes the weights with the participants' credibility level to start the TD process. Moreover, this framework employs two layers of TD to calculate the truth of each group and then the estimated truth of the sensing task. Hence, this framework is mainly divided into two parts; weight initialization and the Two-layer Group-based Truth Discovery (TGTD) mechanism. The former performs TD initial weights initialization with participants' credibility level. The latter further contains weight update and truth update processes within two layers. Finally, the sensing platform presents an accurate estimation of the monitored environment task.

Therefore, the key point in designing this framework is how to calculate TD for a group-based activity for streaming data while preventing the misbehaving participants' data from initializing the truth at the beginning of the TD process. An illustration of the framework is shown in Fig 1, the two steps, weight initialization, and TGTD are explained below.

### Weight initialization

In each time slot $t$, given the set of participants' sensors reading the fog nod and sensing platform calculates the participants' credibility $C_i$ to initialize the weight with the participants'

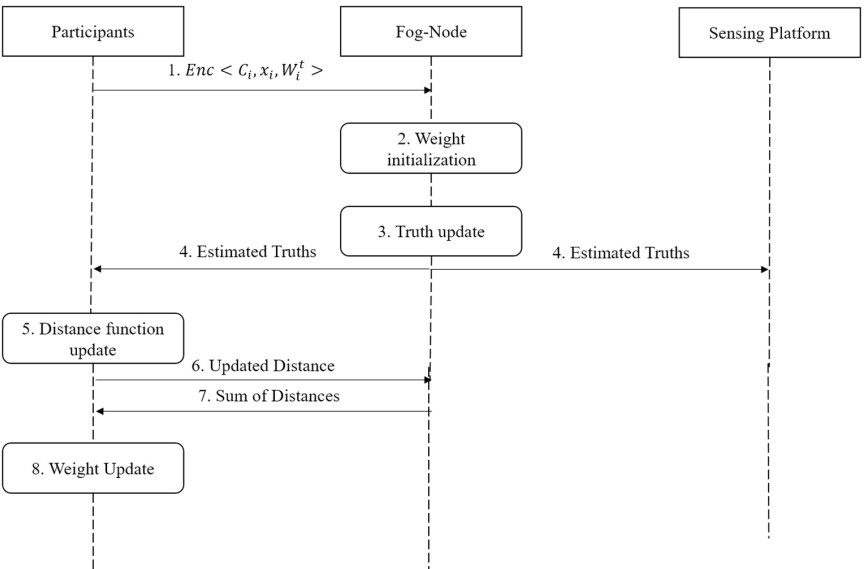

**Fig 1. Framework architecture.**

credibility level. Hence, we make sure the TD process starts with reliable participants' data, and the reliable participants dominate from the start of the TD process.

Here, we adopt the idea at [49], to calculate the participants' credibility, where the participants satisfy multiple factors that represent their readiness and capabilities to perform the sensing task, according to the following equation:

$$C_i = (k_1 \cdot C_{i,old}) + (k_2 \cdot R_i) + (k_3 \cdot Y_i) \tag{5}$$

In this formula, these weight factors $[k_1, k_2, k_3]$, are determined by the requester (i.e. task publisher). They specify the importance of each parameter, such that $k_i \geq 0 \ \forall_i$.

**Participant's old credibility** ($C_{i,old}$)**:** calculated by the sensing platform according to the participant's previous sensing task, initialized with 0 if the participant is new.

**Participant's readiness and capability** ($R_i$)**:** determined by two device-related parameters [50,51], which are computed by the platform when the participant registers in the MCS system, as the following:

$$R_{i \ (E,A)} = \begin{cases} 1 & \text{if } E > \text{predefined battery level} \\ & \quad \text{and } A \text{ available} \\ \\ 0 & \text{otherwise} \end{cases} \tag{6}$$

Where, residual energy ($E$), is a parameter that measures the device battery level, which is updated dynamically during the sensing task. Sensor availability ($A$) is the availability of the required sensors in the participants' mobile devices, we assume that during the registration the platform is aware of each device's sensors.

**Ability ratio** ($Y_i$)**:** is the ratio that measures the ability of the participants to successfully complete the sensing task to the total assigned tasks [50,51]. The set of completed tasks is updated regularly and stored in the sensing platform. The reliability ratio of participants $i$ is given by:

$$Y_i = \frac{|set \ of \ successfully \ completed \ tasks|}{|set \ of \ total \ asiggned \ tasks|} \tag{7}$$

Then, the credibility level computed in Eq 5, is normalized to the range [0,1], as follows:

$$\acute{C}_i = \frac{C_i}{\sum_{i=1}^{|I|} C_i} \tag{8}$$

Where $\acute{C}_i$ is the normalized value of the participant $i's$ credibility level.

Finally, the fog-nod and the sensing platform initialize the initial weights for the TD process with normalized participants' credibility level, as:

$$w_i^0 = \acute{C}_i \tag{9}$$

Hence, we make sure from the beginning of the truth process that only reliable and credible participants contribute to the truth discovery, to reach an accurate estimation of the truth.

## Two-layer Group-based Truth Discovery (TGTD) mechanism

In this mechanism, we consider the calculation of group members' truth and the estimated truth among groups. Each layer contains truth update, distance update, and weight update steps, to finally get the truth $X_S$ for task $S$ Fig 2, gives an overview of the mechanism as a stream of TD, involving two layers of which the first layer of truth discovery works on finding the groups-estimated truth. Then the second layer works on the first-layer group estimated truth to find the final truth $X_S$ for the sensing task.

**The first layer, truth discovery for each group.** This layer performs truth discovery for the group members to find the estimated truth of the groups for the sensing task $S$. Specifically, given members' stream sensor readings $\{x_{S,1}^t, x_{S,2}^t, \dots, x_{S,i}^t\}$, and the weight of each member $\{w_1^t, w_2^t, \dots, w_i^t\}$, the TGTD's first layer calculates an estimated truth of each group $\{X_{1,S}^t, X_{2,S}^t, \dots, X_{G,S}^t\}$, for task $S$ at time slot $t$, according to the following steps:

1. **Truth update for each group**
   Participants submit to the FN their credibility level $C_i$, the weight of the previous time slot $w_{i,}^{t-1}$, and the sensor readings for task $S$ at the time slot $t$, all encrypted to be protected, $Enc \langle C_i,\ w_{i,}^{t-1},\ x_{s,i}^t \rangle$,. Hence, after initializing the weight with the participants' credibility level, the truth updates based on the participants' current sensor readings and the participant's weight of the previous time slot $t$. Therefore, the truth of sensing task $S$ for each group $X_{G,S}^t$ in each time slot $t$, can be estimated as:

$$X_{G,\,S}^t = \frac{\sum_{i=1}^{GZ} w_i^{t-1} * x_{S,\,i}^t}{\sum_{i=1}^{GZ} w_i^{t-1}} \tag{10}$$

2. **Distance update**
   The next step calculates the participant's accumulated distance, where the decay rate, $\alpha$, is adopted to let the most recent sensor readings play more role in the weight update. Here the distance function measures the distance between the participant's reading and

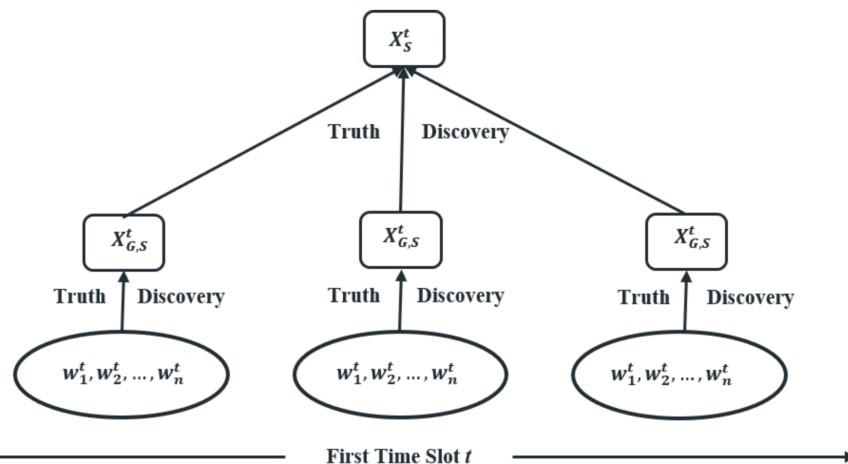

**Fig 2. Overview of TGTD.**

the current group weight to which the participant belongs.

$$D_i^t = D_i^{t-1} \cdot \alpha + D(x_{S,i}^t, X_{G,S}^t) \tag{11}$$

3. **Weight update for each participant**
   The next important function in the TD process is weight update. Therefore, here the weight of each participant is calculated based on the Eq 11, in each time slot $t$. Hence, if the participants' readings are close to the estimated truth of their groups, they are assigned a higher weight:

$$w_i^t = log\left( \frac{\sum_{i=1}^{GZ} D_i^t}{D_i^t} \right) \tag{12}$$

**The second layer, the truth estimation among groups.**

The second layer of TGTD calculates an estimated truth $X_S^t$, based on the groups' estimated truth from the first layers. Specifically, given groups' estimated truth $\{X_{1,S}^t, X_{2,S}^t, \dots, X_{G,S}^t\}$, and the weight of each group $\{w_1^t, w_2^t, \dots, w_G^t\}$, TGTD sets $X_S^t$, as estimated truth for task $S$ for time slot $t$, according to the following steps:

1. **Truth update among groups**
   The truth update is performed between the groups, $\{G_1, G_2, \dots G^*\}$. In this step, the estimates of the truth of task $S$, $X_S^t$, is based on the latest groups' weight estimation, $w_G^{t-1}$, and the group truth estimation, $X_{G,S}^t$, which has been calculated in the first layer.

$$X_S^t = \frac{\sum_{G=1}^{G^*} w_G^{t-1} * X_{G,S}^t}{\sum_{G=1}^{G^*} w_G^{t-1}} \tag{13}$$

2. **Distance update**
   Similar to the first layer, the groups' distance update is based on the distance square function. In this function, we calculate the distance between the groups' estimated truth, $X_{G,S}^t$, computed in the first layer, and the estimated truth for task $S$. Moreover, we also adopt the decay rate to give the most recent groups' truth estimation more role in estimating the final estimated truth.

$$D_G^t = D_G^{t-1} * \alpha + D(X_{G,S}^t, X_S^t) \tag{14}$$

3. **Weight update for each group**
   For each group that participates in the sensing task, their weight is updated based on the distance of the group's estimated truth calculated from the first layer, and the estimated truth. More weight is given to the group as their estimated truth gets closer to the task-estimated truth, which means their sensor reading is closer to the truth value.

$$w_G^t = log\left( \frac{\sum_{G=1}^{G^*} D_G^t}{D_G^t} \right) \tag{15}$$

The TGTD mechanism is designed on Algorithm 2. TGTD has four inputs: the participants' sensor readings, the participants' weight of the previous time slot, the participants'

**Algorithm 2. Two-layer Group-based Truth Discovery (TGTD).**

  **Input:** $Enc < C_i, \ x_i, w_i^{t-1} >, \ decayrate \ \alpha$.

  **Output:** Estimated truth for task $s(X_s^*)$

1: **Initialization:**
2: $w_i^0 \leftarrow \acute{C}_i$ //initialize weight with the normalized Credibility level of participants
3: **End of Initialization:**
4: **while** each time slot **do**
5:   **while** $(G \neq 0)$ **do** //There is still a group
6:     **for** $i = 1$ to $GZ$ **do** //First layer, calculates the truth for each group
7:       Estimate weights with Eq 10
8:       Estimate distance with Eq 11
9:       Estimate truths with Eq 12
10:    **end for** //Second layer, calculates the estimated truth
11:    Estimate weights with Eq 13
12:    Estimate distance with Eq 14
13:    Estimate truths with Eq 15
14:  **end while**
15: **end while**
16: **return:** $X_s^*$

credibility level, and the decay rate. TGTD uses $w_i^0$, to initialize weight with the credibility level of the participants. In each time slot, and for each group TGTD calculates the group's estimated truth $(X_{G,S}^t)$ as a first layer in the TGTD process based on Eqs 10, 11, 12, and lines (7-9). After that, as a second layer, TGTD calculates the final truth $(X_S^t)$, based on Eqs 13, 14, 15, and line (11-16).

The running time for the TGTD algorithm is $O(G)$ for the while loop to go through all the groups (line 5). Then the for-loop to to go through all the group members (line 6), takes $O(i)$. Therefore, TGTD is bounded by $O(G*i)$. Hence, the TGTD algorithm is computationally efficient.

However, the analysis of fog architecture simulation and the organization of streaming processes were beyond the scope of this study, as they do not constitute its primary contributions. These topics are identified as valuable directions for future research and in-depth investigation.

## Performance evaluation

In this section, the TGTD algorithm over the data stream is evaluated by conducting various experiments and simulations on both real-world datasets and Synthetic datasets. We conduct a comparison between our scheme and the baseline framework for incremental truth discovery on streaming data (iCRH) [16], Algorithm (2), and the centralized streaming CATD (Cen.CATD) [41], Algorithm (1). The iCRH proves its state-of-the-art efficiency performance in our target real-time MCS scenario. Moreover, the three approaches utilize iterative methods to infer truth discovery in streaming data. Similar to iCRH and (Cen.CATD), our approach scans readings once per time slot, resulting in fewer computational steps. However, unlike iCRH and (Cen.CATD), which are designed for individual participant scenarios, our approach is tailored for group mobile crowdsensing scenarios.

## Dataset

To demonstrate the effectiveness and efficiency of the proposed TGTD algorithm we use three datasets, two real-world datasets, and one synthetic dataset.

1. **Weather forecast dataset** [33]**:** This dataset contains 18 heterogeneous sources that record daily weather information for 30 cities in the United States, every 45 minutes on a day in Mar 2010 from Jan . 28, 2010 to Feb. 4, 2010. We use the high and low daily temperature properties in the experiments as they are continuous data. Furthermore, we consider the data collected from Accuweather.com as the ground truth.

2. **Stock dataset** [33] **:** This is trading data of 1000 stock symbols collected from 55 sources over 21 working days in July 2011. The volume, shares outstanding, and market cap properties are used in the experiments as they continue data.

    The ground truths are given. Based on the fact that the ground truths are known for both the Weather forecast dataset and the Stock dataset, the weight of the source is quantified by measuring the distances between its reading and the ground truths.

3. **Synthetic dataset:** We generate the synthetic dataset by simulating 50 sources, and sampling random numbers as ground truth. Then we add different levels of Gaussian noise following normal distribution to simulate sensor readings. Furthermore, we divide these sources into 5 groups.

## Performance metrics

To evaluate the framework comprehensively, the following metrics are used:

1. **Accuracy**
   To evaluate the deviation between the estimated results and the real truths. Hence, we measure the resulting accuracy by adopting the standard root of mean squared error (RMSE) [52] of estimated truth against the ground truth, according to:

$$RMSE = \sqrt[2]{\frac{\sum_{t=1}^{T} \sum_{s=1}^{S} \left(X_s^t - X_s^*\right)^2}{T * S}} \tag{16}$$

   The lower the value of RMSE, the better the performance of the scheme. Therefore, to measure if our approach can obtain a more accurate estimation result compared to other approaches. Hence, the smaller the value between the estimated results and the real truths, the higher the score the approaches get.

2. **Efficiency**
   Measured by the running time of the system to show the scalability and efficiency of the framework, the lower the better. Through this matrix, we can see the computing overhead of our approach, which can prove the practicality of the approach.

Finally, we evaluate the effectiveness and efficiency of TGTD by comparing the results of the three algorithms, TGTD, iCRH and (Cen.CATD).

## Simulation setup

In the simulation, to verify the effectiveness and efficiency of the TGTD algorithm we experiment on two real-world datasets and one synthetic dataset and present the result as

follows. All mechanisms are implemented in MATLAB R2022b, and experiments are conducted on a PC equipped with Intel(R) Core (TM) i7-8565U CPU and 16.0 GB RAM, running on Windows 10 (64-bit). Furthermore, to comprehensively evaluate the performance of the TGTD algorithm we vary two parameters, the timestamp and sensing task percentage, to observe the accuracy and efficiency of TGTD with different parameter settings. The participants' credibility is generated randomly following normal distribution in the range [0,1], in different time slots. For simplicity, the population of participants is divided into groups equally. The simulation is conducted on each dataset and the average is taken over 10 runs.

## Evaluation

### *Accuracy*

First, we evaluate the accuracy of the final estimated ground truth by varying the timestamp across three datasets. We use RMSE to measure the deviation between the estimated results and the actual truth. Fig 3 presents the estimation error of TGTD compared to iCRH as a baseline and centralized streaming CATD (Cen.CATD). As shown in Fig 3a, despite some fluctuations in accuracy, TGTD outperforms both the baseline algorithm and (Cen.CATD) on the pedestrian dataset. Similarly, Figures Fig 3b and 3c demonstrate that TGTD achieves higher accuracy than both algorithms on the weather and stock datasets. Although TGTD and iCRH show similar accuracy on the stock dataset, TGTD still has a slightly lower error estimation compared to iCRH. These results indicate that the two-layer TD approach of TGTD enhances the accuracy of the truth discovery process in group-based scenarios. Additionally, TGTD effectively identifies the accurate truth among groups, leading to higher quality sensing tasks.

Fig 4 presents the comparison results of RMSA among TGTD, iCRH, and stream (Cen.CATD), with varying percentages of sensing tasks. As shown in Fig 4a, TGTD exhibits lower error estimation compared to the other two algorithms when the number of tasks increases over the pedestrian dataset. Similarly, Fig 4b and 4c demonstrate that TGTD maintains high accuracy in both the Stock and Weather real-world datasets, outperforming iCRH and stream (Cen.CATD). These results indicate that TGTD remains feasible as the number of sensing tasks increases. TGTD calculates each participant's credibility from the beginning of the truth discovery process, ensuring that only reliable and credible participants contribute to achieving an accurate estimation of the truth. Additionally, the implementation of two layers of truth discovery enhances the accuracy of each sensing task.

In summary, TGTD consistently demonstrates high accuracy across both pedestrian and real-world datasets when varying timestamps and task percentages, compared to the baseline algorithms iCRH and stream (Cen.CATD). These findings highlight the significance of the two layers of truth discovery and the positive impact of initiating the process with credible participants on the overall accuracy of the truth.

### *Efficiency.*

In this section, we evaluate the computation time of truth discovery to demonstrate the efficiency of TGTD. The experimental results in Fig 5 show the increase in computation time across the pedestrian and real-world datasets with varying timestamps. Although TGTD's running time is slightly higher on the pedestrian dataset, it performs nearly as well as the baseline iCRH algorithm, as depicted in Fig 5a. However, it significantly outperforms the streaming Cen.CATD. On the other hand, Fig 5b and 5c clearly show that TGTD has a lower computation time on the Weather and Stock datasets.

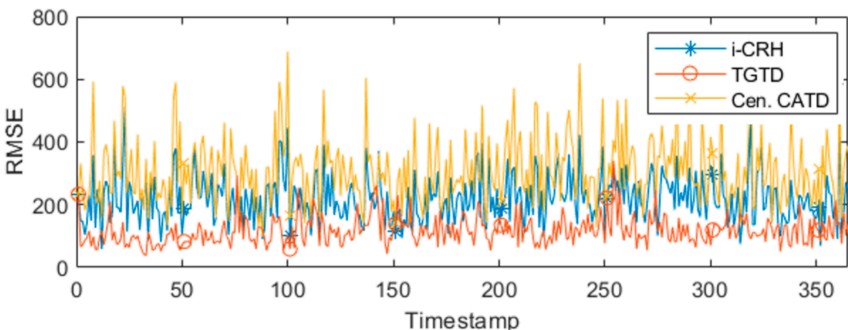

(a) Pedestrian Dataset

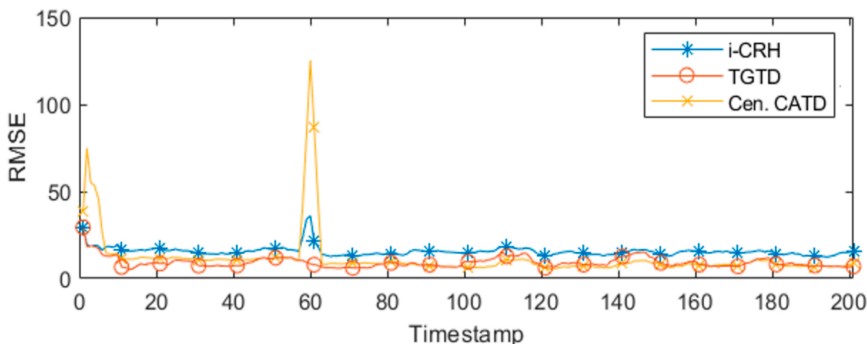

(b) Weather Dataset

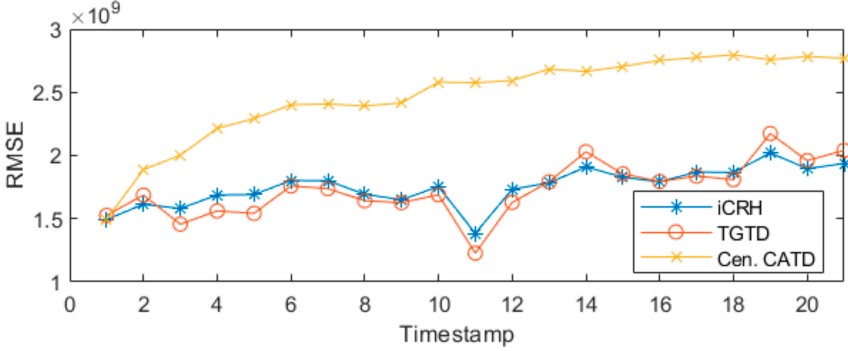

(c) Stock Dataset

**Fig 3. Accuracy evaluation varying timestamp.**

Similarly, Fig 6 illustrates the running time when varying the sensing task percentage. As expected, TGTD's running time is more reasonable compared to streaming Cen.CATD and approaches the running time of the baseline iCRH across all datasets. The slight increase in TGTD's running time is due to the additional computation required to calculate the credibility level of participants as the task percentage increases.

In summary, TGTD takes slightly more time than the baseline iCRH in all cases, but it remains reasonable in real-world scenarios. It consistently outperforms the streamingCen.CATD across all datasets, as shown in Fig 6a, 6b and 6c. One possible explanation

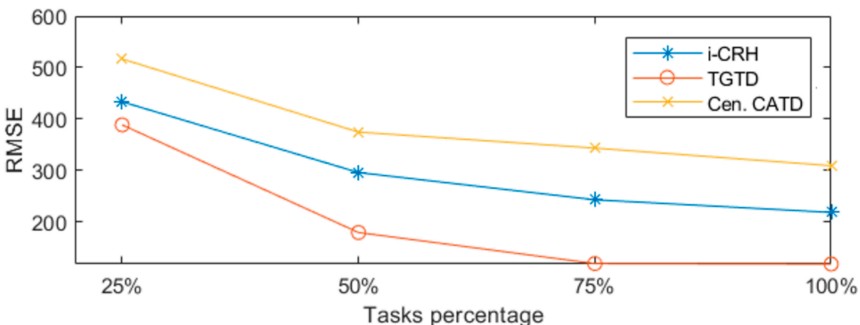

(a) Pedestrian Dataset

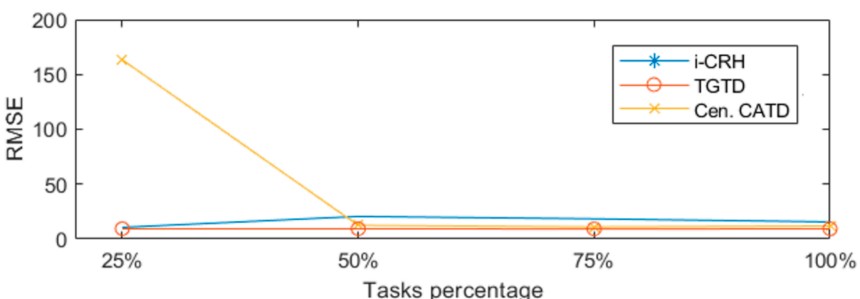

(b) Weather Dataset

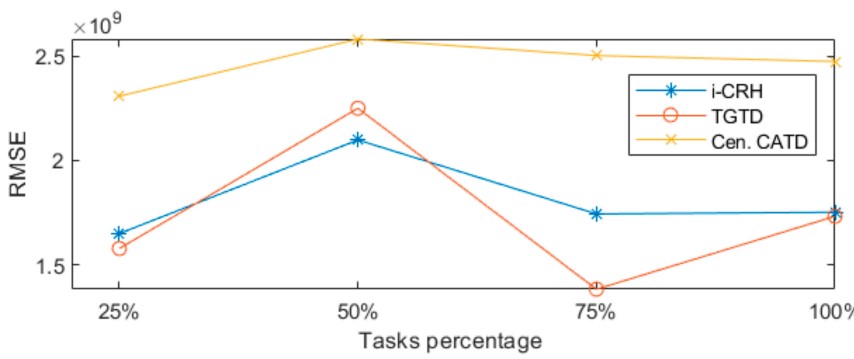

(c) Stock Dataset

**Fig 4. Accuracy evaluation varying task percentage.**

is that TGTD performs two layers of truth discovery, starting with highly credible participants, unlike streaming Cen.CATD, which is more affected by outliers. Nevertheless, TGTD's running time is still acceptable in real-world scenarios, and this is a trade-off for achieving accurate truth discovery for both the group and individual group members after calculating participant credibility.

In conclusion, TGTD is both effective and efficient in comparison to other streaming approaches.

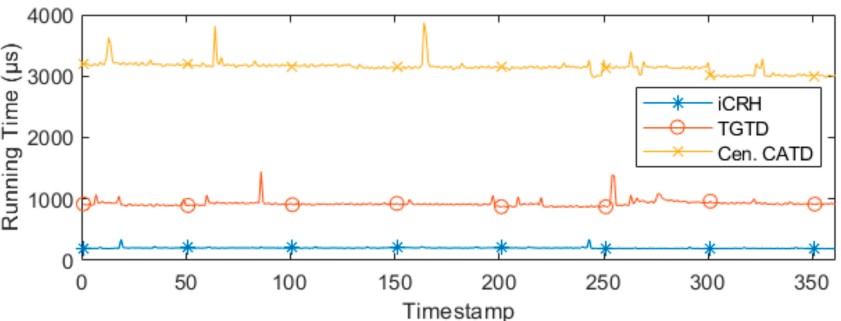

(a) Pedestrian Dataset

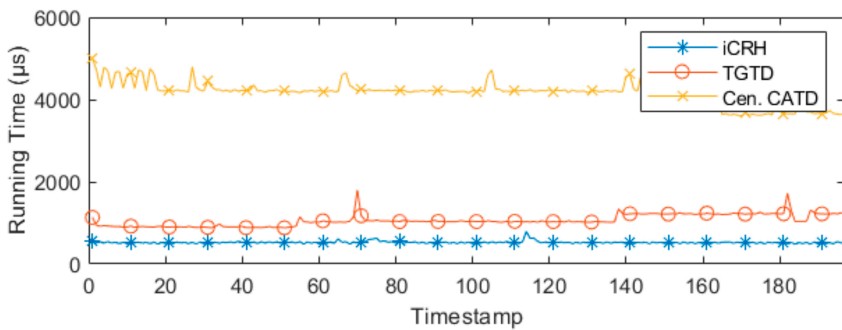

(b) Weather Dataset

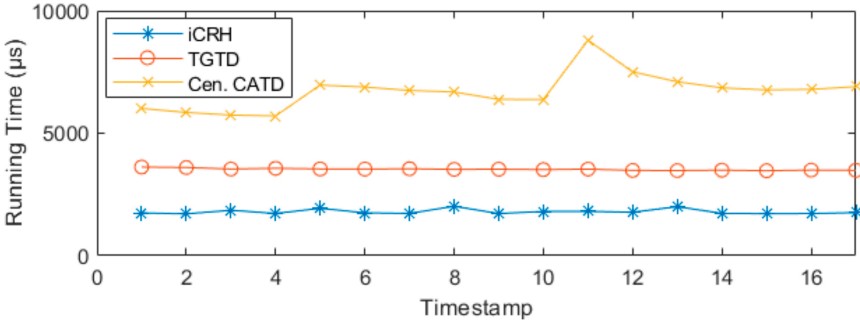

(c) Stock Dataset

**Fig 5. Efficiency evaluation varying timestamp.**

## Conclusion

In this article, we address the problem of group-based TD in the data stream MCS and the problem of misbehaving participants dominating at the start of the truth discovery process. We propose A Fog-Assisted Group-based Truth Discovery Framework over MCS Data Streams. In particular, we developed a novel TD method for crowdsensing data streams where we initialized the weights for the weight update process in TD with participants' credibility level. Where the credibility function is based on the participants' readiness and commitment and the device's ability to perform the sensing tasks. In addition, we develop a novel TGTD mechanism in which the first layer estimates the truth of the group and the second layer estimates the aggregated truth sensing task. Finally, we experimentally evaluate the

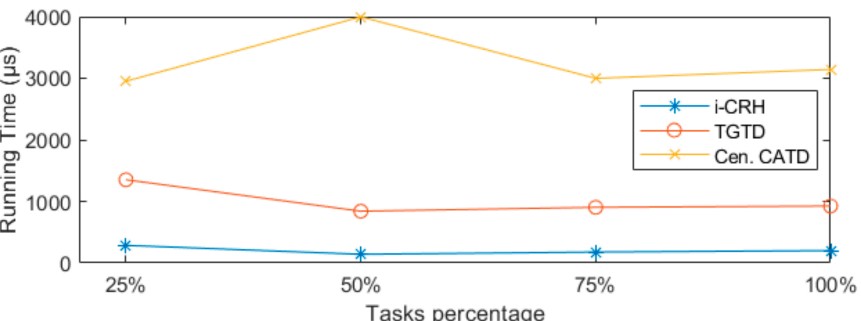

**(a) Pedestrian Dataset**

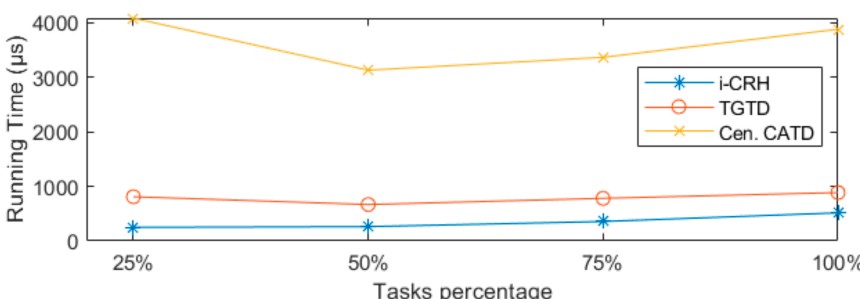

**(b) Weather Dataset**

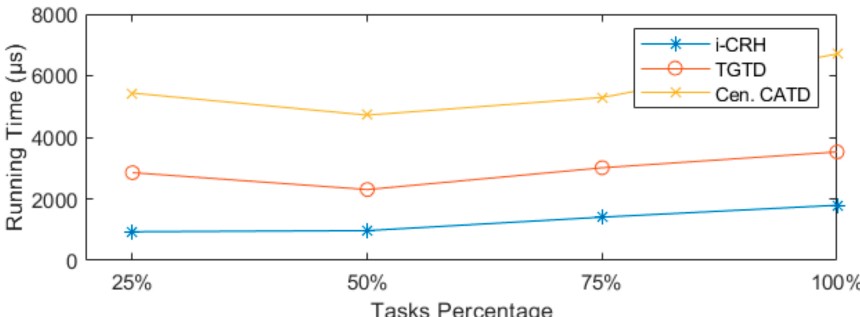

**(c) Stock Dataset**

**Fig 6. Efficiency evaluation varying task percentage.**

effectiveness and efficiency of the TGTD algorithm on both pedestrian and two real-world datasets, comparing it against the baseline iCRH algorithm and the centralized CATD. The results demonstrate that TGTD outperforms both algorithms in terms of accuracy. Although TGTD requires slightly more computation time than the baseline iCRH, it significantly outperforms Cen.CATD. This additional computation time is a trade-off for TGTD's ability to provide accurate truth discovery for group-based activities.

In the future, our aim is to address the organization of the streaming process in the context of fog architecture simulation. In addition, we would like to develop an incentive mechanism for group-based MCS systems. Furthermore, we plan to study the mobility of the participants in the participant recruitment system in MCS.

## Author contributions

**Conceptualization:** Bayan Hashr Saeed Alamri.

**Data curation:** Bayan Hashr Saeed Alamri, Muhammad Mostafa Monowar.

**Formal analysis:** Bayan Hashr Saeed Alamri, Muhammad Mostafa Monowar, Suhair Alshehri.

**Funding acquisition:** Mohammad Haseeb Zafar.

**Investigation:** Bayan Hashr Saeed Alamri, Muhammad Mostafa Monowar, Suhair Alshehri, Mohammad Haseeb Zafar.

**Methodology:** Bayan Hashr Saeed Alamri, Muhammad Mostafa Monowar, Suhair Alshehri.

**Project administration:** Muhammad Mostafa Monowar, Suhair Alshehri, Mohammad Haseeb Zafar.

**Resources:** Suhair Alshehri.

**Software:** Bayan Hashr Saeed Alamri, Muhammad Mostafa Monowar.

**Supervision:** Muhammad Mostafa Monowar, Suhair Alshehri, Mohammad Haseeb Zafar.

**Validation:** Suhair Alshehri, Mohammad Haseeb Zafar.

**Visualization:** Muhammad Mostafa Monowar.

**Writing – original draft:** Bayan Hashr Saeed Alamri.

**Writing – review & editing:** Muhammad Mostafa Monowar, Suhair Alshehri, Mohammad Haseeb Zafar.

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
