## [Decision Letter · Decision Letter 0]

11 Jun 2024

PONE-D-24-14153A Fog-assisted Group-based Truth Discovery Framework over Mobile Crowdsensing Data StreamsPLOS ONE

Dear Dr. Zafar,

Thank you for submitting your manuscript to PLOS ONE. After careful consideration, we feel that it has merit but does not fully meet PLOS ONE’s publication criteria as it currently stands. Therefore, we invite you to submit a revised version of the manuscript that addresses the points raised during the review process.

We look forward to receiving your revised manuscript.

Kind regards,

Fazlullah Khan

Academic Editor

PLOS ONE

Additional Editor Comments:

The authors are requested to address the reviewer's comments thoroughly and submit the revised version soon

Reviewers' comments:

Reviewer's Responses to Questions

**Comments to the Author**

1. Is the manuscript technically sound, and do the data support the conclusions?

Reviewer #1: Yes

Reviewer #2: Yes

Reviewer #3: Yes

2. Has the statistical analysis been performed appropriately and rigorously? 

Reviewer #1: Yes

Reviewer #2: Yes

Reviewer #3: Yes

3. Have the authors made all data underlying the findings in their manuscript fully available?

Reviewer #1: Yes

Reviewer #2: Yes

Reviewer #3: Yes

4. Is the manuscript presented in an intelligible fashion and written in standard English?

Reviewer #1: Yes

Reviewer #2: Yes

Reviewer #3: Yes

5. Review Comments to the Author

Reviewer #1: In this article, the authors address the problem of group-based TD in the data stream MCS and the problem of misbehaving participants dominating at the start of the truth discovery process. In my opinion, this paper contains a lot of advantages and is well organized, however, it has some major limitations and needs to be modified before being accepted.

(1) In the first Introduction, innovation points are proposed to be combined to no more than 3. (2) Detailed analysis is suggested in the Design goals. (3) In each figure containing subgraphs, (()) should be changed to (). (4) The discussion or comparisons with more recent related schemes, such as ppru: a privacy-preserving reputation updating scheme for cloud-assisted vehicular networks, a privacy-preserving and reputation-based truth discovery framework in mobile crowdsensing, a lightweight privacy preservation scheme with efficient reputation management for mobile crowdsensing in vehicular networks,, instead of conventional schemes are suggested. (5) The paper lacks theoretical analysis, and it is suggested to supplement.

To sum up, I think this paper can be accepted if all the above problems are solved well.

Reviewer #2: In this paper, the problem of group-based TD in the data stream MCS and the problem of misbehaving participants dominating at the start of the truth discovery process are addressed. A Fog-Assisted Group-based Truth Discovery Framework over MCS Data Streams is proposed. Here are some comments:

1.What is the basis for selecting the proposed method compared to other methods? Suggested additional explanation.

2.There is still room for improvement in English writing.

Reviewer #3: In this paper, the authors propose a novel Fog-assisted Group-based Truth Discovery Framework over MCS Data Streams, an efficient TD system for real-time applications. Specifically, the authors first initialized the weights for the weight update process in TD with the participants’ credibility level. Then, the authors developed a novel two-layer group-based truth discovery (TGTD) mechanism in which the first layer estimates the truth of the group’s members and the second layer estimates the aggregated truth for the groups. In my opinion, this paper contains a lot of advantages and is well organized, however, it has some major limitations and needs to be modified before being accepted.

(1) In the Introduction section, the “We proposed fog-assisted TD for crowdsensing systems” should be corrected to “We propose fog-assisted TD for crowdsensing systems”. (2) Some explanation should be added to the Design Goals section. (3) The truth discovery process usually exposes the privacy of entities. Can this scheme achieve privacy protection? It is suggested that the author add some discussion on this aspect. (4) The discussion or comparisons with more recent related schemes, such as a privacy-preserving and reputation-based truth discovery framework in mobile crowdsensing, a lightweight privacy preservation scheme with efficient reputation management for mobile crowdsensing in vehicular networks, efficient anonymous authentication and privacy-preserving reliability evaluation for mobile crowdsensing in vehicular networks, lightweight and privacy-preserving dual incentives for mobile crowdsensing, instead of conventional schemes are suggested. (5) “two-layer group-based truth discovery (TGTD) mechanism” should be corrected to “Two-layer group-based truth discovery (TGTD) mechanism”

To sum up, I think this paper can be accepted if all the above problems are solved well.

6. PLOS authors have the option to publish the peer review history of their article (what does this mean?). If published, this will include your full peer review and any attached files.

Reviewer #1: No

Reviewer #2: No

Reviewer #3: No

---

## [Author Response · Author response to Decision Letter 1]

17 Jul 2024

Reply Letter

[Tributes]

We would like to express our deep thanks and gratitude to the reviewers for their comments on the paper. We have tried our best to modify the paper according to their recommendations. We hope our paper will be published in PLOS ONE.

COMMENTS and Answers

Reviewer:1

Comment: 1. In the first Introduction, innovation points are proposed to be combined to no more than 3.

Answer: We appreciate the reviewer’s comment. We re-write the contributions in three points as follows, the “Introduction” section, Page 3.

• “We develop a novel TD method for crowdsensing data streams where we initialized the weights in TD with participants’ credibility level. Then we formulate the credibility function based on the participants’ readiness, commitment, and the device’s ability to perform the sensing tasks. This method guarantees the accuracy of the data as participants with a high credibility level are likely to provide accurate data.

• We develop a novel fog-assisted Two-layer Group-based Truth Discovery (TGTD) mechanism. The first layer estimates the truth of the group’s members and the second layer estimates the aggregated truth for the groups in the area of interest. Additionally, it offloads the process to the FN deployed in a different geographic location, to minimize the overhead on the participants’ side.

• We conduct extensive simulations using synthetic datasets and different real-world datasets and show that the proposed framework outperforms the baseline TD approach.”

Comment: 2 Detailed analysis is suggested in the Design goals.

Answer: Thanks to the reviewer for the comment. Based on the comment, we added these paragraphs to the “Design Goals” Section, Page 8, and the “Performance Metrics” Section, Page 15.

“In this article, we intend to devise a TGTD mechanism, which can provide more accurate ground truth for group sensing activities. Specifically, our mechanism achieves the following twofold design goals.

1. Accuracy: The proposed scheme should output a highly accurate estimation. Accordingly, it is measured by the deviation between the estimated results and the real truths.

2. Efficiency: The proposed scheme should reach a significantly lower overhead for the participants. Thus, it is measured by the running time of the system to show the scalability and efficiency”

“1. Accuracy

To evaluate the deviation between the estimated results and the real truths. Hence, we measure the resulting accuracy by adopting the standard root of mean squared error (RMSE) [50] of estimated truth against the ground truth, according to:

The lower the value of RMSE, the better the performance of the scheme. Therefore, to measure if our approach can obtain a more accurate estimation result compared to other approaches. Hence, the smaller the value between the estimated results and the real truths, the higher the score the approaches get.

2. Efficiency

Measured by the running time of the system to show the scalability and efficiency of the framework, the lower the better. Through this matrix, we can see the computing overhead of our approach, which can prove the practicality of the approach.”

Comment: 3: In each figure containing subgraphs, (()) should be changed to ().

Answer: Thanks to the reviewer for this comment. We recheck the figures and change the format accordingly for all figures, from pages 16 to 19.

Comment: 4: The discussion or comparisons with more recent related schemes, such as ppru: a privacy-preserving reputation updating scheme for cloud-assisted vehicular networks, a privacy-preserving and reputation-based truth discovery framework in mobile crowdsensing, a lightweight privacy preservation scheme with efficient reputation management for mobile crowdsensing in vehicular networks, instead of conventional schemes are suggested.

Answer: Thanks to the reviewer for this comment. We added this paragraph to explain why we compare with the baseline iCRH algorithm. To the Section “Performance Evaluation”, Pages 13, and 14.

“Performance Evaluation

In this section, the TGTD algorithm over the data stream is evaluated by conducting various experiments and simulations on both real-world datasets and Synthetic datasets. We conduct a comparison between our scheme and the baseline framework for incremental truth discovery on streaming data iCRH [16], Algorithm (2). The iCRH proves its state-of-the-art efficiency performance in our target real-time MCS scenario. Similar to iCRH, our approach scans readings once for each time slot and has fewer computation steps. However, compared to iCRH which is for individual participant scenarios, our approach is for group MCS scenarios”.

Note: All the truth discovery frameworks are either based on the CRH algorithm for one-time sensing tasks or iCRH for stream sensing tasks. Hence, as we target stream MCS tasks we compare our algorithm with iCRH.

Comment: 5: The paper lacks theoretical analysis, and it is suggested to supplement.

Answer: We appreciate the comment for theoretical analysis, we added this paragraph at the end of the “Proposed Framework” Section, Page 13.

“The running time for the TGTD algorithm is O(G) for the while loop to go through all the groups (line 5). Then the for-loop to to go through all the group members (line 6), takes O(i). Therefore, TGTD is bounded by O(G ∗ i). Hence, the TGTD algorithm is computationally efficient.”

Note: We prove the efficiency and effectiveness of the framework through experiments on both Synthetic and two real-world datasets.

Reviewer:2

Comment: 1: What is the basis for selecting the proposed method compared to other methods? Suggested additional explanation.

Answer: Thanks to the reviewer for this comment. We added this paragraph to the “Performance Evaluation” Section on page 14.

“The iCRH proves its state-of-the-art efficiency performance in our target real-time MCS scenario. Similar to iCRH, our approach scans readings once for each time slot and has fewer computation steps. However, compared to iCRH which is for individual participant scenarios, our approach is for group MCS scenarios.”

Comment: 2: There is still room for improvement in English writing.

Answer: We appreciate the reviewer’s comment regarding improvement in English writing. We checked and corrected minor typos and long sentences in the revised manuscript.

Reviewer:3

Comment: 1: In the Introduction section, the “We proposed fog-assisted TD for crowdsensing systems” should be corrected to “We propose fog-assisted TD for crowdsensing systems”.

Answer: Thanks to the reviewer for this comment. We corrected the typo in the “Introduction” Section accordingly on page 3.

Comment: 2: Some explanation should be added to the Design Goals section.

Answer: We appreciate the reviewer’s comment. We added these paragraphs to the “Design Goals” Section, Page 8, and the “Performance Metrics” Section, on page 15.

“In this article, we intend to devise a TGTD mechanism, which can provide more accurate ground truth for group sensing activities. Specifically, our mechanism achieves the following twofold design goals.

1. Accuracy: The proposed scheme should output a highly accurate estimation. Accordingly, it is measured by the deviation between the estimated results and the real truths.

2. Efficiency: The proposed scheme should reach a significantly lower overhead for the participants. Thus, it is measured by the running time of the system to show the scalability and efficiency”

“1. Accuracy

To evaluate the deviation between the estimated results and the real truths. Hence, we measure the resulting accuracy by adopting the standard root of mean squared error (RMSE) [50] of estimated truth against the ground truth, according to:

The lower the value of RMSE, the better the performance of the scheme. Therefore, to measure if our approach can obtain a more accurate estimation result compared to other approaches. Hence, the smaller the value between the estimated results and the real truths, the higher the score the approaches get.

2. Efficiency

Measured by the running time of the system to show the scalability and efficiency of the framework, the lower the better. Through this matrix, we can see the computing overhead of our approach, which can prove the practicality of the approach.”

Comment: 3: The truth discovery process usually exposes the privacy of entities. Can this scheme achieve privacy protection? It is suggested that the author add some discussion on this aspect.

Answer: We appreciate the reviewer’s comment regarding the privacy issues, but as we mentioned at the end of the “Related work” Section, Pages 6 and 7, the following statement.

“In addition, we adopt a non-private version of the iCRH in this work.”

Hence, the privacy concern is out of the scope of this paper.

Comment: 4: The discussion or comparisons with more recent related schemes, such as a privacy-preserving and reputation-based truth discovery framework in mobile crowdsensing, a lightweight privacy preservation scheme with efficient reputation management for mobile crowdsensing in vehicular networks, efficient anonymous authentication and privacy-preserving reliability evaluation for mobile crowdsensing in vehicular networks, lightweight and privacy-preserving dual incentives for mobile crowdsensing, instead of conventional schemes are suggested.

Answer: Thanks to the reviewer for this comment. We added this paragraph to explain why we compare with the baseline iCRH algorithm. To the Section “Performance Evaluation”, Pages 13, and 14.

“Performance Evaluation

In this section, the TGTD algorithm over the data stream is evaluated by conducting various experiments and simulations on both real-world datasets and Synthetic datasets. We conduct a comparison between our scheme and the baseline framework for incremental truth discovery on streaming data iCRH [16], Algorithm (2). The iCRH proves its state-of-the-art efficiency performance in our target real-time MCS scenario. Similar to iCRH, our approach scans readings once for each time slot and has fewer computation steps. However, compared to iCRH which is for individual participant scenarios, our approach is for group MCS scenarios”

Note: All the truth discovery frameworks are either based on the CRH algorithm for one-time sensing tasks or iCRH for the stream sensing task. Hence, as we target stream MCS tasks we compare our algorithm with iCRH.

Comment: 5: two-layer group-based truth discovery (TGTD) mechanism” should be corrected to “Two-layer group-based truth discovery (TGTD) mechanism”.

Answer: Thanks to the reviewer for this comment. We rechecked the whole paper and corrected it to the “Two-layer Group-based Truth Discovery (TGTD) mechanism” accordingly.

---

## [Decision Letter · Decision Letter 1]

29 Sep 2024

PONE-D-24-14153R1A Fog-assisted Group-based Truth Discovery Framework over Mobile Crowdsensing Data StreamsPLOS ONE

Dear Dr. Zafar,

Thank you for submitting your manuscript to PLOS ONE. After careful consideration, we feel that it has merit but does not fully meet PLOS ONE’s publication criteria as it currently stands. Therefore, we invite you to submit a revised version of the manuscript that addresses the points raised during the review process.

We look forward to receiving your revised manuscript.

Kind regards,

Fazlullah Khan

Academic Editor

PLOS ONE

Additional Editor Comments:

According to the reviewers' comments and non-serous attitude toward addressing comments the paper should be rejected (the reviewers think to reject the paper). However, the editorial office wants to give a chance to the authors to rerun the simulation and compare their result with the lasted paper such as few list below. If the comments are not fully addresses, the paper will be rejected automatically.

(1) A Lightweight Privacy Preservation Scheme with Efficient Reputation Management for Mobile Crowdsensing in Vehicular Networks, IEEE TDSC, 10.1109/TDSC.2022.3163752

(2) A Privacy-Preserving and Reputation-Based Truth Discovery Framework in Mobile Crowdsensing, IEEE TDSC, 10.1109/TDSC.2023.3276976

(3) Efficient Anonymous Authentication and Privacy-Preserving Reliability Evaluation for Mobile Crowdsensing in Vehicular Networks, IEEE IOTJ, 10.1109/JIOT.2023.3259961

(4) Lightweight and Privacy-Preserving Dual Incentives for Mobile Crowdsensing, IEEE TCC, 10.1109/TCC.2024.3372598

Reviewers' comments:

Reviewer's Responses to Questions

**Comments to the Author**

1. If the authors have adequately addressed your comments raised in a previous round of review and you feel that this manuscript is now acceptable for publication, you may indicate that here to bypass the “Comments to the Author” section, enter your conflict of interest statement in the “Confidential to Editor” section, and submit your "Accept" recommendation.

Reviewer #1: (No Response)

Reviewer #3: (No Response)

2. Is the manuscript technically sound, and do the data support the conclusions?

Reviewer #1: Partly

Reviewer #3: Partly

3. Has the statistical analysis been performed appropriately and rigorously? 

Reviewer #1: N/A

Reviewer #3: No

4. Have the authors made all data underlying the findings in their manuscript fully available?

Reviewer #1: Yes

Reviewer #3: Yes

5. Is the manuscript presented in an intelligible fashion and written in standard English?

Reviewer #1: Yes

Reviewer #3: Yes

6. Review Comments to the Author

Reviewer #1: It seems that the author did not revise the paper seriously, so it is suggested that the author follow the reviewer's opinion and make careful revisions before submitting the paper for review again.

Reviewer #3: The author did not revise it carefully according to the last comment. The comparison scheme is that of 2016, and the experiment is not convincing. The authors should add a discussion or comparison with the latest options listed below to make your paper more convincing.

(1) A Lightweight Privacy Preservation Scheme with Efficient Reputation Management for Mobile Crowdsensing in Vehicular Networks, IEEE TDSC, 10.1109/TDSC.2022.3163752

(2) A Privacy-Preserving and Reputation-Based Truth Discovery Framework in Mobile Crowdsensing, IEEE TDSC, 10.1109/TDSC.2023.3276976

(3) Efficient Anonymous Authentication and Privacy-Preserving Reliability Evaluation for Mobile Crowdsensing in Vehicular Networks, IEEE IOTJ, 10.1109/JIOT.2023.3259961

(4) Lightweight and Privacy-Preserving Dual Incentives for Mobile Crowdsensing, IEEE TCC, 10.1109/TCC.2024.3372598

7. PLOS authors have the option to publish the peer review history of their article (what does this mean?). If published, this will include your full peer review and any attached files.

Reviewer #1: No

Reviewer #3: No

---

## [Author Response · Author response to Decision Letter 2]

2 Nov 2024

Reply Letter

[Tributes]

We would like to express our deep thanks and gratitude to the reviewers for their comments on the paper. We have tried our best to modify the paper according to their recommendations. We hope our paper will be published in PLOS ONE.

COMMENTS and Answers

Reviewer:1

Comment: 1. It is suggested that the author follow the reviewer's opinion and make careful revisions before submitting the paper for review again.

Answer: Thank you for your feedback. We apologize if our previous revisions did not fully address the reviewer’s comments. We have now carefully reviewed and revised the paper, ensuring that all suggestions and concerns have been thoroughly addressed. All the changes have been highlighted in Yellow. We appreciate your patience and look forward to your further feedback on our revised submission.

Reviewer:3

Comment: 1. The authors should add a discussion or comparison with the latest options listed below to make your paper more convincing.

Answer: We appreciate the reviewer’s comment. In response, we have re-experimented and compared our approach with the latest streaming truth discovery approach, Cen-CATD [1] in addition to the baseline approach [2]. Based on the reviewer’s feedback, we have revised the Evaluation section, which now spans pages 16 to 21.

Note: The list of papers mentioned in the comments does not address dynamic data truth discovery algorithms (i.e., they do not consider timestamps of the data, but rather focus on convergence criteria). Both iCRH and our algorithm, TGTD, iterate over each timestamp of streaming/dynamic data, rather than using a loop with convergence-stopping criteria. Therefore, comparing algorithms that rely on timestamps with those that use convergence criteria is not fair. To ensure a fair comparison, we have identified the latest streaming truth discovery algorithms and re-experimented them, comparing our algorithm with two other streaming truth discovery approaches.

[1] P. S. Mukkamala, H. Wu, and B. Düdder, "Reliable and Streaming Truth Discovery in Blockchain-based Crowdsourcing," in 2023 20th Annual IEEE International Conference on Sensing, Communication, and Networking (SECON), 2023, pp. 492-500: IEEE.

[2] Y. Li et al., "Conflicts to harmony: A framework for resolving conflicts in heterogeneous data by truth discovery," IEEE Transactions on Knowledge and Data Engineering, vol. 28, no. 8, pp. 1986-1999, 2016.

---

## [Decision Letter · Decision Letter 2]

31 Mar 2025

PONE-D-24-14153R2A Fog-assisted Group-based Truth Discovery Framework over Mobile Crowdsensing Data StreamsPLOS ONE

Dear Dr. Zafar,

Thank you for submitting your manuscript to PLOS ONE. After careful consideration, we feel that it has merit but does not fully meet PLOS ONE’s publication criteria as it currently stands. Therefore, we invite you to submit a revised version of the manuscript that addresses the points raised during the review process.

We look forward to receiving your revised manuscript.

Kind regards,

Muhammad Anwar, Ph.D.

Academic Editor

PLOS ONE

Reviewers' comments:

Reviewer's Responses to Questions

**Comments to the Author**

1. If the authors have adequately addressed your comments raised in a previous round of review and you feel that this manuscript is now acceptable for publication, you may indicate that here to bypass the “Comments to the Author” section, enter your conflict of interest statement in the “Confidential to Editor” section, and submit your "Accept" recommendation.

Reviewer #3: (No Response)

Reviewer #4: (No Response)

2. Is the manuscript technically sound, and do the data support the conclusions?

Reviewer #3: (No Response)

Reviewer #4: Yes

3. Has the statistical analysis been performed appropriately and rigorously? 

Reviewer #3: (No Response)

Reviewer #4: N/A

4. Have the authors made all data underlying the findings in their manuscript fully available?

Reviewer #3: (No Response)

Reviewer #4: Yes

5. Is the manuscript presented in an intelligible fashion and written in standard English?

Reviewer #3: (No Response)

Reviewer #4: Yes

6. Review Comments to the Author

Reviewer #3: In this paper, the authors propose a novel Fog-assisted Group-based Truth Discovery Framework over MCS Data Streams, an efficient TD system for real-time applications. Specifically, the authors first initialized the weights for the weight update process in TD with the participants’ credibility level. Then, the authors developed a novel two-layer group-based truth discovery (TGTD) mechanism in which the first layer estimates the truth of the group’s members and the second layer estimates the aggregated truth for the groups. In my opinion, this paper contains a lot of advantages and is well organized, however, it has some major limitations and needs to be modified before being accepted.

(1) In the Introduction section, the “We proposed fog-assisted TD for crowdsensing systems” should be corrected to “We propose fog-assisted TD for crowdsensing systems”. (2) Some explanation should be added to the Design Goals section. (3) The truth discovery process usually exposes the privacy of entities. Can this scheme achieve privacy protection? It is suggested that the author add some discussion on this aspect. (4) The discussion or comparisons with more recent related schemes, such as a privacy-preserving and reputation-based truth discovery framework in mobile crowdsensing, a lightweight privacy preservation scheme with efficient reputation management for mobile crowdsensing in vehicular networks, efficient anonymous authentication and privacy-preserving reliability evaluation for mobile crowdsensing in vehicular networks, lightweight and privacy-preserving dual incentives for mobile crowdsensing, instead of conventional schemes are suggested. (5) “two-layer group-based truth discovery (TGTD) mechanism” should be corrected to “Two-layer group-based truth discovery (TGTD) mechanism”

To sum up, I think this paper can be accepted if all the above problems are solved well.

Reviewer #4: 1.There is no clear evidence or explanation of how the authors simulate the fog architecture, including latency, sensing platform, the number of fog nodes, participants, and requesters, in addition to how the components are distributed as well as the techniques used to transfer data between them. Without such a setup, the current work cannot be considered related to fog computing.

2.There is no explanation of how the streaming process is organized and how the sensors/ data producers are simulated.

3.In the evaluation section, it is unclear how the fog node contributes to performance improvement.

4.Please ensure that all abbreviations are properly introduced by mentioning their full terms upon first use. Kindly review all abbreviations.

5.There is a claim that certain papers assume all participants and workers are reliable. However, some of these papers may not have explicitly discussed the issue of reliability. This claim requires further clarification.

6.In Fig. 1: Framework Architecture, it would be beneficial to add and distinguish the Requester. Additionally, please explain why the weight update and distance function are not part of the fog node.

7.There is no discussion regarding the scalability of the proposed framework, nor is there an explanation of its limitations.

8.Please improve the quality of the figures.

7. PLOS authors have the option to publish the peer review history of their article (what does this mean?). If published, this will include your full peer review and any attached files.

Reviewer #3: No

Reviewer #4: No

---

## [Author Response · Author response to Decision Letter 3]

9 Apr 2025

We would like to express our deep thanks and gratitude to the reviewers for their comments on the paper. We have tried our best to modify the paper according to their recommendations. We hope our paper will be published in PLOS ONE.

COMMENTS and Answers

Reviewer:3

Comment: 1. In the Introduction section, the “We proposed fog-assisted TD for crowdsensing systems” should be corrected to “We propose fog-assisted TD for crowdsensing systems”.

Answer: We corrected the typo in the “Introduction” Section accordingly.

Comment: 2. Some explanation should be added to the Design Goals section.

Answer: Thanks to the reviewer for the comment. Based on the comment, we added these paragraphs to the “Design Goals” Section, Page 8, and the “Performance Metrics” Section, Page 15.

“In this article, we intend to devise a TGTD mechanism, which can provide more accurate ground truth for group sensing activities. Specifically, our mechanism achieves the following twofold design goals.

1. Accuracy: The proposed scheme should output a highly accurate estimation. Accordingly, it is measured by the deviation between the estimated results and the real truths.

2. Efficiency: The proposed scheme should reach a significantly lower overhead for the participants. Thus, it is measured by the running time of the system to show the scalability and efficiency”

“1. Accuracy

To evaluate the deviation between the estimated results and the real truths. Hence, we measure the resulting accuracy by adopting the standard root of mean squared error (RMSE) [50] of estimated truth against the ground truth, according to:

The lower the value of RMSE, the better the performance of the scheme. Therefore, to measure if our approach can obtain a more accurate estimation result compared to other approaches. Hence, the smaller the value between the estimated results and the real truths, the higher the score the approaches get.

2. Efficiency

Measured by the running time of the system to show the scalability and efficiency of the framework, the lower the better. Through this matrix, we can see the computing overhead of our approach, which can prove the practicality of the approach.”

Comment: 3. The truth discovery process usually exposes the privacy of entities. Can this scheme achieve privacy protection?

Answer: We sincerely appreciate the reviewer's comment regarding privacy issues. As indicated at the end of the "Related Work" section on 7, we have stated:

"In addition, we adopt a non-private version of the iCRH in this work."

Therefore, privacy concerns fall outside the scope of this paper.

Comment: 4. The discussion or comparisons with more recent related schemes, such as a privacy-preserving and reputation-based truth discovery framework in mobile crowdsensing, a lightweight privacy preservation scheme with efficient reputation management for mobile crowdsensing in vehicular networks, efficient anonymous authentication and privacy-preserving reliability evaluation for mobile crowdsensing in vehicular networks, lightweight and privacy-preserving dual incentives for mobile crowdsensing, instead of conventional schemes are suggested.

Answer: We appreciate the reviewer's comment. In response, we have re-experimented and compared our approach with the latest streaming truth discovery approach, Cen-CATD [1] (reference [41] in the paper), in addition to the baseline approach [2] (reference [16] in the paper). Based on the reviewer's feedback, we have revised the Evaluation section, which now spans pages 16 to 21.

Please note that the list of papers mentioned in the comments does not address dynamic data truth discovery algorithms (i.e., they do not consider timestamps of the data, but rather focus on convergence criteria). Both iCRH and our algorithm, TGTD, iterate over each timestamp of streaming/dynamic data, rather than using a loop with convergence-stopping criteria. Therefore, comparing algorithms that rely on timestamps with those that use convergence criteria is not fair. To ensure a fair comparison, we have identified the latest streaming truth discovery algorithms and re-experimented them, comparing our algorithm with two other streaming truth discovery approaches.

Comment: 5. “two-layer group-based truth discovery (TGTD) mechanism” should be corrected to “Two-layer group-based truth discovery (TGTD) mechanism”.

Answer: We thank the reviewer for their insightful comment. We have thoroughly rechecked the entire paper and have made the necessary corrections to reflect the "Two-layer Group-based Truth Discovery (TGTD) mechanism" accordingly.

Reviewer:4

Comment: 1. There is no clear evidence or explanation of how the authors simulate the fog architecture, including latency, sensing platform, the number of fog nodes, participants, and requesters, in addition to how the components are distributed as well as the techniques used to transfer data between them.

Answer: We sincerely appreciate the reviewer's detailed feedback. Our research primarily focuses on truth discovery in MCS, which is why we did not delve into the simulation aspects of the fog architecture, including latency, sensing platform, the number of fog nodes, participants, requesters, component distribution, and data transfer techniques. We believe that addressing these aspects would significantly extend the length of the article. However, we acknowledge the importance of these elements and suggest that future research could explore these areas in depth.

Comment: 2. There is no explanation of how the streaming process is organized and how the sensors/ data producers are simulated.

Answer: We appreciate the reviewer's insightful comments. Our research primarily focuses on truth discovery in MCS. Given that most MCS applications continuously collect streaming data, with distributions that are often unknown and constantly changing throughout the data collection process, we did not provide a detailed explanation of the organization of the streaming process or the simulation of sensors/data producers. We believe that addressing these aspects would extend the scope of our current study beyond its intended focus.

Comment: 3. In the evaluation section, it is unclear how the fog node contributes to performance improvement.

Answer: We thank the reviewer for their insightful comments. Our evaluation section primarily focuses on the accuracy and efficiency of our truth discovery mechanism, comparing it with others in the field. This is why we did not detail how the fog node contributes to performance improvement. We believe that including this information would extend the scope of our current study beyond its intended focus. However, we acknowledge the importance of understanding the fog node's role in performance improvement and suggest that future research could explore this aspect in greater detail.

Comment: 4. Please ensure that all abbreviations are properly introduced by mentioning their full terms upon first use. Kindly review all abbreviations.

Answer: We appreciate the reviewer's feedback. In response, we have redefined all abbreviations, and highlighted the changes in yellow.

Comment: 5. There is a claim that certain papers assume all participants and workers are reliable. However, some of these papers may not have explicitly discussed the issue of reliability. This claim requires further clarification.

Answer: Thank you for the insightful comment. In response, we have added the following paragraph to the “Introduction” section on Page 2:

“They either rely on the observation that the majority of users contribute reliable data even in the absence of ground truth [8, 11]. Some methods initialize a random ground truth at the beginning of the TD process [18], assuming the worker weight to be known [19, 23], or assign random weights [20–22] at the start”.

Comment: 6. In Fig. 1: Framework Architecture, it would be beneficial to add and distinguish the Requester. Additionally, please explain why the weight update and distance function are not part of the fog node.

Answer: Thank you for your valuable feedback regarding Fig. 1: Framework Architecture. We appreciate your suggestion to add and distinguish the Requester. However, our primary focus is on the accuracy of the collected sensing data by participants and its proximity to the ground truth, which is why we have not included the Requester in the diagram.

Each fog node manages a group of participants and facilitates communication between them and the platform. After the fog node initializes the TD process, each participant calculates the distance between their sensory data and the ground truth. The summed distances are then sent to all participants for the weight update. This approach ensures that the weight update and distance calculation processes are distributed among participants, rather than being centralized within the fog node.

Comment: 7. There is no discussion regarding the scalability of the proposed framework, nor is there an explanation of its limitations.

Answer: Thank you for your valuable feedback. I appreciate your insights regarding the scalability and limitations of the proposed framework.

Regarding scalability, our system employs a group approach for participants, which inherently supports scalability. Additionaly, in the evaluation section, we varied the task percentage to its maximum to demonstrate that the accuracy and efficiency of our framework remain higher compared to other approaches.

As for limitations, we have mentioned in the conclusion that we plan to study the mobility of participants in truth discovery in MCS. This aspect significantly impacts truth discovery and is an area we intend to explore further

Comment: 8. Please improve the quality of the figures.

Answer: Thank you for your valuable comment. Based on your feedback, we have enhanced the quality of all the figures in the article to ensure clarity and precision.

---

## [Decision Letter · Decision Letter 3]

11 Jul 2025

PONE-D-24-14153R3A Fog-assisted Group-based Truth Discovery Framework over Mobile Crowdsensing Data StreamsPLOS ONE

Dear Dr. Zafar,

Thank you for submitting your manuscript to PLOS ONE. After careful consideration, we feel that it has merit but does not fully meet PLOS ONE’s publication criteria as it currently stands. Therefore, we invite you to submit a revised version of the manuscript that addresses the points raised during the review process.

We look forward to receiving your revised manuscript.

Kind regards,

Muhammad Anwar, Ph.D.

Academic Editor

PLOS ONE

Journal Requirements:

Reviewers' comments:

Reviewer's Responses to Questions

**Comments to the Author**

1. If the authors have adequately addressed your comments raised in a previous round of review and you feel that this manuscript is now acceptable for publication, you may indicate that here to bypass the “Comments to the Author” section, enter your conflict of interest statement in the “Confidential to Editor” section, and submit your "Accept" recommendation.

Reviewer #4: (No Response)

Reviewer #5: All comments have been addressed

2. Is the manuscript technically sound, and do the data support the conclusions?

Reviewer #4: Yes

Reviewer #5: Yes

3. Has the statistical analysis been performed appropriately and rigorously? 

Reviewer #4: Yes

Reviewer #5: N/A

4. Have the authors made all data underlying the findings in their manuscript fully available?

Reviewer #4: Yes

Reviewer #5: Yes

5. Is the manuscript presented in an intelligible fashion and written in standard English?

Reviewer #4: Yes

Reviewer #5: Yes

6. Review Comments to the Author

Reviewer #4: The authors’ responses to my comments clarify that fog architecture simulation and streaming process organisation are not their primary focus, which aligns with my observation that these elements may be overemphasised. The authors suggest that future research could explore fog architecture and participant mobility, indicating that they view fog computing as an area for further investigation rather than a fully developed component of their current work. I could see that the TGTD mechanism and credibility-based weight initialisation are the primary innovations. These could be implemented without mentioning fog computing (e.g., on a centralised server-client or cloud).

Again, these concepts of fog computing and streaming organisation are relevant and necessary for the MCS context, but are not the primary contributions. While the proposed work is valuable and I would recommend for publication, but, the authors should clearly state in the abstract and in the Proposed Framework section that fog architecture simulation, streaming process organisation are not their primary contributions and have not been fully analysed, and these areas of research are areas for further investigation.

Reviewer #5: The manuscript have been revised and it is more suitable for publication to the best of my knowledge

7. PLOS authors have the option to publish the peer review history of their article (what does this mean?). If published, this will include your full peer review and any attached files.

Reviewer #4: No

Reviewer #5: **Yes: **Sunday Ajagbe

---

## [Author Response · Author response to Decision Letter 4]

12 Jul 2025

[Tributes]

We would like to express our deep thanks and gratitude to the reviewers for their comments on the paper. We have tried our best to modify the paper according to their recommendations. We hope our paper will be published in PLOS ONE.

COMMENTS and Answers

Reviewer#4

Comment: 1. While the proposed work is valuable and I would recommend for publication, but, the authors should clearly state in the abstract and in the Proposed Framework section that fog architecture simulation, streaming process organisation are not their primary contributions and have not been fully analysed, and these areas of research are areas for further investigation.

Answer: We thank the reviewer for the valuable comment. In response, we have revised the abstract to include the following statement: 'The organization of the streaming process within the fog architecture simulation is identified as an area for further investigation and future work.'

Moreover, in the Proposed Framework section, we stated that ‘However, the analysis of fog architecture simulation and the organization of streaming processes were beyond the scope of this study, as they do not constitute its primary contributions. These topics are identified as valuable directions for future research and in-depth investigation.’

Additionally, we have updated the conclusion by expanding the future work section to state: 'In future work, we aim to address the organization of the streaming process in the context of fog architecture simulation.'

Reviewer#5

Comment: 1. The manuscript has been revised, and it is more suitable for publication to the best of my knowledge.

Answer: Thank you very much, we sincerely appreciate that.

---

## [Decision Letter · Decision Letter 4]

5 Aug 2025

A Fog-assisted Group-based Truth Discovery Framework over Mobile Crowdsensing Data Streams

PONE-D-24-14153R4

Dear Dr. Zafar,

We’re pleased to inform you that your manuscript has been judged scientifically suitable for publication and will be formally accepted for publication once it meets all outstanding technical requirements.

Kind regards,

Muhammad Anwar, Ph.D.

Academic Editor

PLOS ONE

Additional Editor Comments (optional):

Reviewers' comments:

Reviewer's Responses to Questions

**Comments to the Author**

1. If the authors have adequately addressed your comments raised in a previous round of review and you feel that this manuscript is now acceptable for publication, you may indicate that here to bypass the “Comments to the Author” section, enter your conflict of interest statement in the “Confidential to Editor” section, and submit your "Accept" recommendation.

Reviewer #4: All comments have been addressed

2. Is the manuscript technically sound, and do the data support the conclusions?

Reviewer #4: Yes

3. Has the statistical analysis been performed appropriately and rigorously? 

Reviewer #4: N/A

4. Have the authors made all data underlying the findings in their manuscript fully available?

Reviewer #4: Yes

5. Is the manuscript presented in an intelligible fashion and written in standard English?

Reviewer #4: (No Response)

6. Review Comments to the Author

Reviewer #4: (No Response)

7. PLOS authors have the option to publish the peer review history of their article (what does this mean?). If published, this will include your full peer review and any attached files.

Reviewer #4: No

---

## [Editor Report · Acceptance letter]

PONE-D-24-14153R4

PLOS ONE

Dear Dr. Zafar,

I'm pleased to inform you that your manuscript has been deemed suitable for publication in PLOS ONE. Congratulations! Your manuscript is now being handed over to our production team.

Kind regards,

on behalf of

Dr. Muhammad Anwar

Academic Editor

PLOS ONE